# *FgFAD12* Regulates Vegetative Growth, Pathogenicity and Linoleic Acid Biosynthesis in *Fusarium graminearum*

**DOI:** 10.3390/jof10040288

**Published:** 2024-04-14

**Authors:** Yimei Zhang, Zhen Gao, Yinyu Lei, Liuye Song, Weijie He, Jingrong Liu, Mengge Song, Yafeng Dai, Guang Yang, Andong Gong

**Affiliations:** 1College of Life Science, Xinyang Normal University, Xinyang 464000, China; 15236758136@163.com (Z.G.); yinyulei2022@163.com (Y.L.); 18436262373@163.com (L.S.); liujingrong0826@163.com (J.L.); 18211736285@163.com (M.S.); daiyafeng0515@163.com (Y.D.); guangyangxynu@126.com (G.Y.); 2Henan Key Laboratory of Tea Plant Biology, Xinyang 464000, China; 3College of Plant Science and Technology, Huazhong Agricultura University, Wuhan 430070, China; heweijie@webmail.hzau.edu.cn

**Keywords:** *Fusarium graminearum*, fatty acid desaturase, *FgFAD12*, pathogenicity

## Abstract

Polyunsaturated fatty acids (PUFAs), as important components of lipids, play indispensable roles in the development of all organisms. ∆12 fatty acid desaturase (*FAD12*) is a speed-determining step in the biosynthesis of PUFAs. Here, we report the characterization of *FAD12* in *Fusarium graminearum,* which is the prevalent agent of Fusarium head blight, a destructive plant disease worldwide. The results demonstrated that deletion of the *FgFAD12* gene resulted in defects in vegetative growth, conidial germination and plant pathogenesis but not sexual reproduction. A fatty acid analysis further proved that the deletion of *FgFAD12* restrained the reaction of oleic acid to linoleic acid, and a large amount of oleic acid was detected in the cells. Moreover, the ∆*Fgfad12* mutant showed increased resistance to osmotic stress and reduced tolerance to oxidative stress. The expression of *FgFAD12* did show a temperature-dependent manner, which was not affected at a low temperature of 10 °C when compared to 25 °C. RNA-seq analysis further demonstrated that most genes enriched in fatty acid metabolism, the biosynthesis of unsaturated fatty acids, fatty acid biosynthesis, fatty acid degradation, steroid biosynthesis and fatty acid elongation pathways were significantly up-regulated in the ∆*Fgfad12* mutants. Overall, our results indicate that *FgFAD12* is essential for linoleic acid biosynthesis and plays an important role in the infection process of *F. graminearum*.

## 1. Introduction

Fusarium head blight of wheat, caused by the fungal pathogen *Fusarium graminearum*, leads to severe devastating food security and economic loss [1,2]. *F. graminearum* also invades barley and maize, leading to crown rot or stalk rot [3]. In addition to reducing the yield and quality of the grain, *F. graminearum* also produced several types of mycotoxins, such as type B trichothecenes deoxynivallenol (DON), zearalenone (ZEN) and nivalenol (NIV) during the infection process [4]. DON inhibits protein synthesis in eukaryotic organisms, which is harmful to humans and animals [5,6,7]. Due to its destructive impact on agriculture, *F. graminearum* is considered one of the top 10 most important fungal plant pathogens worldwide [8]. However, plant diseases are difficult to be controlled in practice because of lacking effective plant resistance cultivars to *F. graminearum* [9]. Therefore, new approaches to cellular and molecular mechanisms to control *F. graminearum* have been studied using genomics, transcriptomics and reverse genetics during the last decade [10,11,12,13,14,15,16]. These studies enrich our knowledge of hyphal growth, conidia production and pathogenicity in *F. graminearum*.

Fatty acids are important components of biofilms, which are responsible for receiving signals from external environmental factors [17]. The changes in the physiological, biochemical and physical properties of the biofilm directly reflect the impact of the environment on organisms. The composition and content of fatty acids are closely related to biological stress resistance [18]. Fatty acids can be divided into saturated fatty acids and unsaturated fatty acids. Two or more double bonds in straight-chain fatty acids containing 18~22 carbon atoms are called polyunsaturated fatty acids (PUFAs). PUFAs mainly contain four groups, including n-3, n-6, n-7 and n-9, that are named based on the position of the first double bond in a carbon chain. The groups of n-3 and n-6 PUFAs play important roles in biological function, such as alpha-linolenic acid (ALA, n-3), eicosapentaenoic acid (EPA, n-3), docosahexaenoic acid (DHA, n-3), linoleic acid (LA, n-6), gamma-linolenic acid (GLA, n-6), and arachidonic acid (ARA, n-6). DHA is a precursor of anti-inflammatory docosanoids and EPA is the substrate for the synthesis of anti-inflammatory eicosanoids. ARA is the precursor of prostaglandins, prostacyclins and leukotrienes involved in inflammatory reactions and immune response [19]. Additionally, PUFAs play a significant role in regulating cell membrane structure, fluidity, signal transduction, material transport, ion exchange, energy transfer and enzyme activity [20,21]. Considering their biological functions, PUFAs are considered key factors in health and disease. PUFAs, especially DHA and EPA, produced from deep-sea fish were popular with humans. However, the reduction of fish resources, the complexity of mining and the increase of marine pollution have become the limiting factors in n-3 PUFA production [22]. Microbial lipids have attracted attention to the production of PUFAs, considering their rich sources. Owing to the easy cultivation, short growth cycle, low production cost and high yield of lipids, oleaginous filamentous fungi have become the main source in PUFA production, and the fungal fermentation of PUFAs has become an international development trend [23,24]. Therefore, analyzing the synthetic pathway and regulation mechanism of PUFAs is conducive to achieving a high quantity of high-quality PUFAs by fungi fermentation technology.

In fungi, under the reaction of different fatty acid desaturases and elongases, the synthesis of PUFAs begins with the dehydrogenation of 18 carbon stearic acid (STA, C18:0) and carbon chain elongation. Various fatty acid desaturases were identified, which can insert the double bond into the carbon chain at a unique position. Delta-5 desaturase (Δ5), Delta-6 desaturase (Δ6), Delta-9 desaturase (Δ9), Delta-12 desaturase (Δ12) and Delta-15 desaturase (Δ15) are well-known fatty acid desaturases. Δ12 desaturase, also known as oleate desaturase, has the capability to introduce a double bond between the 12th and 13th carbon atom counted from the carboxyl group of monounsaturated oleic acid to form polyunsaturated linoleic acid containing two double bonds [25]. Linoleic acid has the physiological effect of reducing the content of cholesterol and triglycerides in blood, preventing Atherosclerosis [26]. Linoleic acid is also the precursor of ALA, GLA, EPA and some physiological regulators such as prostaglandin [26]. The Δ12 desaturase gene of *Mortierella alpine* was cloned and expressed in *Saccharomyces cerevisiae*, leading to the production of linoleic acid at levels up to 25% of the total fatty acids [27]. Potatoes contained higher amounts of PUFAs and a higher tolerance to low temperatures and oxidative stress when they were transformed with the Δ12 desaturase gene from *Cyanobacterium synechocystis* [28]. The Δ12 desaturase gene from *Mortierella isabellina* was successfully expressed in yeast, tobacco, rapeseed and soybean, which significantly increased the content of fatty acids. In vitro, adding oleic acid and linoleic acid could promote the mycelial growth and asexual spore production of *Aspergillus flavus*, while adding saturated fatty acid such as palmitic acid had no effect [29]. Similarly, the addition of EPA could promote the production of asexual spores of *Aspergillus fumigatus* and *Aspergillus nidulans* [30], indicating that unsaturated fatty acids can regulate the growth and development of fungi.

Δ12 desaturase is the key enzyme for the dehydrogenation of oleic acid to form linoleic acid and a rate-limiting enzyme to synthesize n-3 and n-6 PUFAs metabolic pathways, controlling the production of most PUFAs in organisms. Orthologs of Δ12 desaturase are well conserved, but their roles in plant pathogenic fungi have not been studied during pathogenesis. In this study, we aimed to determine the functions of the *FgFAD12* gene in *F. graminearum*. The ∆*Fgfad12* mutant was generated and analyzed for its defects in growth, stress resistance, pathogenesis and LA biosynthesis. In addition, qRT-PCR assay showed that a low temperature stimulated Δ9 and Δ15 fatty acid desaturase gene expression.

## 2. Materials and Methods

### 2.1. Fungal Strains and Culture Conditions

The first table in Section 3.2 lists all strains used in this study. All strains were generally cultured on Potato Dextrose Agar (PDA) plates at 25 °C [31]. Colony morphology and growth rate of mutants were measured on Complete Medium (CM) plates at 25 °C for three days. Conidia were obtained in liquid carboxy-methl-cellulose (CMC) medium at 25 °C with shaking at 175 rpm [32]. Conidiation was measured after 5 days by hemocytometer. For stress response assays, strains were cultured on CM plates with 0.7 M NaCl, 0.05% H_2_O_2_, 0.01% SDS or 300 μg/mL Congo Red [31]. The characters were recorded 5 dpi.

Sexual reproduction was assayed on carrot agar according to Zheng’s method [33]. Protoplast preparation and polyethylene glycol-mediated fungal transformation were performed as described [32]. Final concentrations of 300 µg/mL hygromycin B (CalBiochem, La Jolla, CA, USA) and 400 µg/mL geneticin (Sigma-Aldrich, St. Louis, MO, USA) were used for transformant selection. 

### 2.2. Generation of Fgfad12 Deletion Mutants

To generate the ∆*Fgfad12* mutants, hygromycin phosphotransferase fragments were fused with the upstream and downstream flanking fragments of FgFAD12 gene by overlapping PCR [34]. The upstream and downstream fragments and hygromycin phosphotransferase fragments were amplified with Fad12/1F-2R, Fad12/3F-4R, and HYG/F-R, respectively. The recombination DNA fragments were transformed into the protoplasts of the wild-type PH-1 [35]. The ∆*Fgfad12* mutants were screened by PCR with primer pairs Fad12/5F-6R, H850-H852, Fad12/7F-H855R and H856F-Fad12/R. All primers used for PCR are listed in Appendix A.

### 2.3. Generation of the FgFAD12-GFP and Co-Localization Transformants

*FgFAD12*-GFP fusion construct was generated using the ClonExpress^®^ II One Step Cloning Kit (Vazyme, Nanjing, China). To generate the *FAD12*-GFP fusion construct, the *FAD12* gene with its native promoter was amplified with primer pair Fad12/HF-HR (Appendix A). Subsequently, the *FAD12* gene fragments were cloned into the pKNT-GFP vector by *Hind* III/*Kpn* I double-digested to obtain *FAD12*-GFP fusion construct [36]. The resulting *FgFAD12*-GFP fusion construct was transformed into ∆*Fgfad12* mutants. Transformants containing hygromycin and genecin resistance were verified by PCR, and GFP signals were observed with epifluorescence microscope (Nikon, DS-Ri1, Tokyo, Japan).

For co-localization transformants, FgKar2-mCherry-T-vector was obtained from Shuli Cao of Jiangsu Academy of Agricultural Sciences. FgKar2-mCherry and Fad12-GFP were co-transformed into the wild-type PH-1. Transformants were examined GFP and mCherry signals with epifluorescence microscope (Nikon, DS-Ri1, Tokyo, Japan).

### 2.4. Plant Infection Assays and DON Production Assays

For flowering wheat head infection assays, conidia of PH-1 and mutant strains harvested from 5-day-old CMC cultures were resuspended to 2 × 10^5^ spores/mL in sterile distilled water. The flowering wheat heads of cultivar Annong8455 were inoculated with 10 μL of conidia suspension at the fifth spikelet from the base of spike, as previously described [37]. The inoculated wheat heads were examined at 14 days post-inoculation (dpi) to estimate the disease index [38].

For wheat coleoptile infection assays, germ-free scissors were used to remove 1 mm of the tip of the coleoptile, and then cotton drenched with 2 × 10^5^ spores/mL conidia suspension was wrapped around the wound site [39]. Cotton was removed at 2 dpi, and the length of coleoptile stalks infected was measured at 10 dpi. The experiment was conducted three times, and 10 plants were inoculated in each treatment every time.

For corn silks infection assays, fresh corn silks were inoculated with 7 mm blocks of PH-1, and mutant strains were cultured in the dark. The length of corn silks infected was measured at 7 dpi [40].

For DON production assays, 6 g sterile rice was infected with 1 × 10^5^ spores/mL conidia suspension, then fully mixed the rice and conidia. The infected rice was incubated in the dark for 21 days and regularly stirred to prevent clumping. DON production was measured as described [38].

### 2.5. qRT-PCR Assays

For detecting the expression of ∆9, ∆12 and ∆15 fatty acid desaturase genes, RNA samples were isolated from vegetative hyphae harvested from PDA plates at 10 °C and 25 °C for 6 days with the TRIzol reagent (Invitrogen, Carlsbad, CA, USA). cDNA was synthesized with the M5 Super plus qPCR RT kit with gDNA remover (Beijing, Mei5, China). *FgTUB2* was used as internal control [41], and relative expression levels of each gene were calculated by 2^−∆∆Ct^ method [42]. Data from three biological replicates were used to calculate the mean and standard deviation.

### 2.6. Fatty Acids Analysis

PH-1 and ∆*FgFad12* mutant strains conidia were harvested from CMC cultures 5 dpi, resuspended with Yeast Extract Peptone Dextrose Medium (YEPD) and cultured at 25 °C with shaking at 175 rpm. Vegetative hyphae harvested after 24 h were frozen drying in vacuum freeze dryer.

Samples were mixed with 100 μL 5 mg/mL C17 fatty acid methyl ester/petroleum ether solution, 2 mL 5% concentrated sulfuric acid/methanol solution and 300 μL toluene, mixing slightly, and mixture was kept at 95 °C for 1 h in order to trans-esterify lipids. Adding 2 mL 0.9% NaCl and extract with 1 mL n-hexane to mixture was centrifuged at 5000 rpm for 5 min. Fatty acid content of supernatant was detected by gas chromatography (Agilent 7890A, Santa Clara, CA, USA) equipped with flame ionization detector (FID) using DB FastFAME chromatographic column. Fatty acids were identified by comparison of retention times to authentic standards. The relative amounts of fatty acid were determined by peak areas in the total ion chromatograms. Data from three biological replicates were used to calculate the mean and standard deviation.

### 2.7. RNA-seq Analysis

Vegetative hyphae of PH-1 and ∆*FgFad12* mutants were obtained from 18 h YEPD cultures. Total RNA samples were extracted with TRIzol Reagent (Invitrogen, Carlsbad, CA, USA), and mRNA enrichments were used with Oligo (dT) Beads (Thermo Fisher Scientific, Olso, Norway). Strand-specific RNA-seq libraries were prepared with Truseq^TM^ RNA sample prep Kit (San Diego, CA, USA) and sequenced with Illumina Novaseq6000 (San Diego, CA, USA) with a 2 × 150 bp paired-end read mode at the Origingene Bio-pharm Technology firm (Shanghai, China). For each library, at least 20 Mb of paired-end reads were obtained.

The reference genomic sequence of *F. graminearum* strain PH-1 [43] was used from https://fungi.ensembl.org/Fusarium_graminearum/Info/Index (accessed on 22 June 2023). Reads were mapped onto the reference genome using HISAT2 (version 2.1) [44]. The number of reads aligned to each gene was calculated by RSeQC (version 3.0.1). Differentially expressed genes were analyzed with DESeq2 (version 1.20). Genes with an FDR < 0.05 and log2 fold-change > 1 were considered to be significantly different expressed genes. RNA-seq mappings of all samples were used for assembling transcripts by StringTie [45], and the merged transcript GTF files were applied to the reference gene annotation for CASH. RNA-seq data with GSA number CRA012955 was deposited in National Genomics Data Center.

## 3. Results

### 3.1. FgFAD12 Is a Delta-12 Fatty Acid Desaturase in F. graminearum

Delta-12 (∆12) fatty acid desaturase regulates the conversion of oleic acid to linoleic acid. To identify the ∆12 fatty acid desaturase in *F. graminearum*, PAS_chr4_0052 of *Pichia pastoris* GS115 as a query sequence was used in a BLAST search in the *F. graminearum* protein database. FGSG_05784, one putative ∆12 fatty acid desaturase, was identified, named FgFad12. In different species, fatty acid desaturases contain three conserved histone-rich regions. Protein sequence analysis revealed that the amino acid of FgFad12 was conserved in fungi containing HXXXH, HXXHH, and HXXHH sequence features (Appendix A). A bioinformatics analysis showed that FgFad12 contained 475 amino acid residues, and the predicted molecular weight was approximately 53.41 kDa. Four transmembrane helices were distributed in FgFad12 and analyzed by TMHMM-2.0 (Figure 1A).

Phylogenetic analysis of amino acid sequences was performed with the Fad12 family in different eukaryotic organisms (Figure 1B). This result showed that FgFad12 orthologs widely exist in filamentous fungi, and FgFad12 has a closer genetic relationship with *Fusarium* species.

### 3.2. FgFAD12 Is Required for Vegetative Growth and Conidial Germination, but Not Sexual Reproduction

To explore the function of FgFad12 in *F. graminearum*, we generated the *FgFAD12* deletion mutant via homologous recombination. All mutants had similar phenotypes, although only data for *Fgfad12* mutant ∆*Fgfad12* was presented below (Table 1). When the wild-type strain PH-1, *Fgfad12* mutant ∆*FgFad12* and the complementary strain ∆*FgFad12-C* were cultured on PDA and 5×YEG plates at 25 °C for 3.5 days, the growth rate of *Fgfad12* mutant was significantly reduced compared with PH-1 and ∆*FgFad12-C* (Figure 2A,B; Table 2). The asexual reproduction of the ∆*Fgfad12* was detected in a CMC medium cultured for 5 days. In comparison with PH-1 and the complementary strain, the conidiation of the *Fgfad12* mutant was not reduced (Table 2), suggesting that the deletion of *FgFAD12* does not affect conidiation in *F. graminearum*. However, the conidial germination of the ∆*Fgfad12* was significantly reduced, observed at 4 h and 8 h hyphae cultured in YEPD medium (Figure 2C).

Sexual reproduction plays a critical role in the infection cycle of *F. graminearum*. To analyze the effect of the *FgFAD12* gene on the sexual reproduction of *F. graminearum*, we cultured the *Fgfad12* mutant on carrot plates at 10 days post-fertilization or even longer to observe the perithecium and ascospores formation. The ∆*FgFad12* mutant produced a similar number of perithecia and normal ascopores in comparison with PH-1 (Figure 2D). These results indicate that *FgFAD12* plays an important role in vegetative growth conidia germination but is dispensable when it comes to sexual reproduction.

### 3.3. FgFAD12 Is Involved in Various Stress Responses

To examine the stress response of ∆*Fgfad12* mutant, all strains were cultured on CM plates with 0.7 mol/L NaCl, 0.05% H_2_O_2_, 300 μg/mL Congo red and 0.01% SDS, respectively. The results showed that the inhibition rate of ∆*Fgfad12* mutant was lower than PH-1 and ∆*Fgfad12-C* strains on CM medium with NaCl, Congo red and SDS (Figure 3A,B), indicating that cell membrane and cell wall integrity are increased in ∆*Fgfad12* mutant. Compared with PH-1 and ∆*Fgfad12-C*, the percentage inhibition of ∆*Fgfad12* mutant was increased on CM medium with H_2_O_2_ (Figure 3A,B), indicating that ∆*Fgfad12* mutant decreases the tolerance to oxidative stress. Taken together, these results indicate that *FgFAD12* is involved in response to hyperosmotic conditions, cell destabilization and oxidative stress.

### 3.4. FgFAD12 Is Essential for Pathogen Virulence

To investigate the role of *FgFAD12* in plant infection, flowering wheat heads were point inoculated with conidial suspensions of PH-1, ∆*FgFad12* and ∆*FgFad12*-*C* strains. At 14 days post-inoculation (dpi), the average disease index of the *Fgfad12* mutant was 3.2 (Table 2). Compared with PH-1 15.2 and ∆*FgFad12-C* 14.6, the virulence of *Fgfad12* mutant was significantly reduced (Figure 4A,D, Table 2). To confirm this result, culture blocks of the same strains inoculated wheat coleoptiles and corn silks. Consistent with the mycelial growth results, the *Fgfad12* mutant also showed attenuated virulence as compared with PH-1 and ∆*Fad12-C* (Figure 4B,C,E,F, Table 2). These results implied that *FgFAD12* plays an essential role in plant infection of *F. graminearum*.

DON is an important virulence factor for *F. graminearum* infection. We also assayed DON production in the infected rice grain cultures. In comparison with the wild type, the ∆*FgFad12* mutant was significantly reduced in DON production (Table 2), suggesting that deletion of *FgFAD12* affects DON biosynthesis in *F. graminearum*.

### 3.5. Subcellular Localization of FgFad12 to the ER

For complementation assays, the *FgFAD12*-GFP fusion construct was generated and transformed into the ∆*FgFad12* mutant strain. The resulting complemented transformant ∆*Fad12*-*C* had the wild-type phenotype (Figure 2A), indicating that the functions of FgFad12 were not affected by fusion with GFP protein and the deletion of *FgFAD12* is directly responsible for the defects observed in the ∆FgFad12 mutant. When examined by epifluorescence microscopy, GFP signals were observed only in the endoplasmic reticulum (ER) (Figure 5). ER localization of FgFad12 was consistent with Euk-mPLoc 2.0 (http://www.Csbi-o.sjtu.edu.cn/cgi-bin/EukmPLoc2.cgi, accessed on 12 September 2023) [47]. To confirm this observation, we co-transformed the *FgFAD12*-GFP and *FgKAR2*-mCherry into the wild-type strain PH-1. In the resulting transformant CL3, both GFP and mCherry signals were observed in the ER of hyphae (Figure 5). These results indicated that FgFad12 localizes to the ER.

### 3.6. Low Temperature Affects the Expression of Fatty Acid Desaturases

To assay the growth rate of the ∆*Fgfad12* mutant at low temperature, PH-1, ∆*FgFad12* and ∆*FgFad12*-*C* were cultured on PDA plates at 10 °C for 10 days. Compared with the wild type PH-1, reduced by 50.6% at 25 °C, the ∆*FgFad12* mutant was reduced by approximately 51.5% in growth rate when cultured in 10 °C condition (Figure 6A,B). These results indicate that the *FgFAD12* gene is dispensable for low temperatures in *F. graminearum*.

PUFA content is a crucial factor in enhancing membrane fluidity for organism adaptation to low temperatures [48,49]. *FAD9*, *FAD12* and *FAD15* fatty acid desaturases are important enzymes in the biosynthesis of PUFAs [50]. To determine the expression level of these genes in low-temperature conditions, RNA samples were isolated from 6 days of vegetative hyphae of the wild-type PH-1 cultured on 10 °C and 25 °C PDA plates. Compared with 25 °C, the expression levels of *FgFAD9* and *FgFAD15* were up-regulated over 8- and 23-fold, respectively, in 10 °C growth conditions (Figure 6C). However, the expression level of *FAD12* was not changed (Figure 6C). These results reveal that low temperature induces the expression of fatty acid desaturase genes to increase PUFA content.

### 3.7. ∆Fgfad12 Mutant Is Defective in LA Formation

To determine whether the deletion of the *FgFAD12* gene affected fatty acid content, 24 h hyphae of the wild-type PH-1 and ∆*Fgfad12* mutant cultured at 25 °C were harvested for detection. Fatty acids with chain lengths of 10, 16 and 18 carbons in their saturated and unsaturated forms were detected, the C18 ones accounting for about 87% and 93% in PH-1 and ∆*Fgfad12* mutants, respectively. Fatty acids content of C10:0, C16:0 and C18:0 was similar between PH-1 and ∆*Fgfad12* mutant strains. In the ∆*Fgfad12* strain, the deletion of *FgFAD12* led to a lack of linoleic acid (C18:2), while oleic acid (C18:1) was highly accumulated (Table 3). Linolenic acid of ∆*Fgfad12* mutant accounted for 1.96% was declined severely in comparison with 25.82% of the wild type PH-1. The desaturation pathway was almost blocked because of the *FgFAD12* deletion. These results showed that the catalysis from oleic acid to linoleic acid is mainly responsible for *FgFAD12* in *F. graminearum*.

### 3.8. FgFad12 Affects Unsaturated Fatty Acid Biosynthesis

To explore the effects on gene transcription, we performed RNA-seq analysis with RNA isolated from 18 h hyphae of PH-1 and ∆*Fgfad12* mutant collected from YEPD cultures. Compared with PH-1, 1611 differentially expressed genes (721 up-regulated and 890 down-regulated) were detected in the ∆*Fgfad12* mutant (Figure 7A), accounting for 17.1% of total expressed genes. Gene Ontology (Go) terms enrichment analysis was performed on differentially expressed genes. Most down-regulated genes play roles in cytoplasmic translation, organonitrogen compound metabolic process, cellular biosynthetic process, ribosomal subunit biogenesis and ribosome assembly (Figure 7B). However, up-regulated genes were enriched in different Go terms. In the biological process category, genes were enriched in the cellular lipid metabolic process, fatty acid metabolic process, fatty acid biosynthetic process, lipid biosynthetic process and sterol biosynthetic process (Figure 7C). In the molecular function category, fatty acid synthase activity, fatty-acyl-CoA synthase activity and acetate-CoA ligase activity were changed significantly in the ∆*Fgfad12* mutant (Figure 7C). 

KEGG pathway analysis was performed to reveal gene function in the signaling pathway. Among the down-regulated genes, 78 genes were involved in 60 and 40 subunits biosynthesis of ribosome (Figure 7D). Other down-regulated genes were related to starch and sucrose metabolism, amino sugar and nucleotide sugar metabolism, galactose metabolism and carotenoid biosynthesis (Figure 7D). In the up-regulated genes group, fatty acid metabolism was the top pathway, followed by biosynthesis of unsaturated fatty acids, citrate cycle, fatty acid degradation and steroid biosynthesis (Figure 7E). Sterols are essential membrane components and regulate membrane permeability, fluidity, stability, aerobic metabolism and the activities of membrane-bound enzymes [51,52]. Ergosterol is a fungal-specific sterol and is the major sterol of the plasma membrane [53]. A total of 8 genes involved in ergosterol biosynthesis were increased significantly in the ∆*Fgfad12* mutant. To analyze the ergosterol content of PH-1 and ∆*Fgfad12* strains, 5 days of aerial hyphae of the wild-type PH-1 and ∆*Fgfad12* mutant cultured on PDA plates were harvested. The ergosterol content of 87.42 × 10^−4^% of the ∆*Fgfad12* mutant was higher than the PH-1 ergosterol content of 60.98 × 10^−4^% (Appendix A). This result indicated that the deletion of the *FgFAD12* gene stimulates the biosynthesis of ergosterol, which supports the RNA-seq data. RNA-seq analysis and ergosterol content assay demonstrated that the ∆*Fgfad12* mutant improves the production of ergosterol by increasing the expression levels of the key enzymes of the sterol synthesis pathway, thereby enhancing the integrity of the plasma membrane.

## 4. Discussion

As a receptor for organisms, cell membranes can directly respond to physiological and biochemical changes and quickly adapt to adverse conditions. The biological activity of cell membranes relies on the lipid bilayer composed of membrane lipids and membrane proteins and the biofilm fluidity [54]. Fatty acids, especially PUFAs, are the major structural components of cell membranes. Through changing the membrane composition, fatty acids provide various biological effects to regulate transcription and cellular signaling [55]. Δ12 fatty acid desaturase plays a key role in PUFA biosynthesis. In this study, we first presented the functional characterization of ∆12 fatty acid desaturase (FgFad12) in *F. graminearum*.

The deletion of *FgFAD12* resulted in a significantly reduced growth rate and conidia germination but is dispensable for conidiation and sexual reproduction. Fad12 orthologs are conserved in ascomycetes, but the functions of this gene in oleaginous yeast and pathogenic filamentous fungi were not exactly the same. In *Yarrowia lipolytica*, the growth of the ∆fad2 strain was comparable to the wild-type strain at 30 °C on the YNB medium containing various carbon sources [56]. In *Aspergillus parasiticus*, the growth of the Δ^12^-desaturase mutant was decreased twofold. Moreover, spore germination, conidiation and sclerotial development were also inhibited compared to wild-type strains [57]. In *Aspergillus nidulans,* the ∆*odeA* (∆12 desaturase) deletion mutant was reduced in conidial production and vegetative growth, even though the ratio of asexual spores to ascospores is affected [58]. The exogenous addition of linoleic acid can promote mycelial growth and asexual spore production in *Aspergillus flavus*. These results indicate that ∆12 fatty acid desaturase genes with conserved sequences and structures showed varied biological functions in different fungi.

In the stress sensitivity, compared with PH-1, the ∆*Fgfad12* mutant was more sensitive to H_2_O_2_. This result is consistent with Peyou-Ndi’s research. The transformed *Saccharomyces cereviase* strain showed higher resistance to ethanol and oxidative when the heterologously expressed ∆12 fatty acid desaturase gene was isolated from *Caenorhabditis elegans* [59]. Interestingly, the ∆*Fgfad12* mutant was more tolerant to SDS and CR, indicating that the deletion of *FgFAD12* enhances the cell membrane and cell wall integrity. The deletion of *FgFAD12* blocked PUFA biosynthesis, thereby decreasing the cell membrane fluidity and enhancing the rigidity of the cell membrane. The increased ergosterol content of the ∆*Fgfad12* mutant also enhanced the rigidity of the cell membrane. The increased tolerance to SDS may relate to the changes in the rigidity of the cell membrane. Ergosterol is also involved in the maintenance of cell wall integrity. The increased ergosterol leads to the remodeling of the cell wall, making it more difficult for chitin and β-1,3-glucans on the cell wall to be exposed, ultimately making the ∆*Fgfad12* mutant insensitive to CR [60]. Compared with the 17% growth inhibition of PH-1, the ∆*Fgfad12* mutant had 4% growth promotion when cultured with 0.7 mol/L NaCl, suggesting that the deletion of the *FgFAD12* gene makes mutants more adaptable to a high-permeability environment. The composition and content of fatty acids affect the functions of cell membranes and walls. In *Pichia pastoris*, the deletion of *FAD9A* caused a decrease in plasma membrane unsaturation levels and an increase in the tolerance to SDS and hyperosmotic stress [61]. When *Rhizopus* R31.6 is cultured on a 2% salt medium, in unsaturated fatty acid (UFA), including oleic acid, the content of linoleic acid and linoleic acid are all significantly reduced [62]. The physical properties of plasma membranes, such as fluidity and rigidity, are determined by the lipid composition of biofilms [63]. In other words, increased UFA levels raise the membrane fluidity, and conversely, decreased UFA levels lead to enhanced rigidity. In addition, RNA-seq analysis displayed that the expression levels of eight genes (FGSG_06215, FGSG_04994, FGSG_02502, FGSG_02783, FGSG_05740, FGSG_01000, FGSG_04092 and FGSG_09830) related to ergosterol biosynthesis were prominently increased in the mutant strains compared with wild type. Ergosterol extraction assay revealed that the ergosterol content in the ∆*Fgfad12* mutant is 1.4 times higher than the wild-type PH-1 (Appendix A). Ergosterol is abundant in fungal plasma membranes and plays an important role in plasma membrane integrity, fluidity and permeability [64,65]. The membrane integrity of *the* plasma membrane may be rescued by increased production of ergosterol in the ∆*Fgfad12* mutant.

Wheat head, wheat coleoptile and corn silk infection assays showed that the *FgFAD12* gene plays a critical role in plant infection. The ∆*Fgfad12* mutant reduced growth rate and had increased sensitivity to H_2_O_2_ (Figure 2B and Figure 3B), which may contribute to its defects in plant infection. Fatty acid analysis showed that the deletion of the *FgFAD12* gene barely produced PUFAs in *F. graminearum* (Table 3). PUFAs are essential for the growth and virulence of pathogens [66,67,68]. In addition, according to RNA-seq data, decreased expression levels of genes in the ∆*Fgfad12* mutant, including FGSG_02279, FGSG_04745 and FGSG_05042, played roles in plant infection [49,69,70]. The reduced expression of these genes may also contribute to defects of ∆*Fgfad12* mutant in plant infection.

In the present study, we found that the expression levels of ∆9 and ∆15 desaturases were substantially increased, but ∆12 desaturases had no obvious change when *F. graminearum* was cultivated at 10 °C compared with that at 25 °C. ∆9 is the only desaturase existing in all types of organisms, which play an indispensable role in maintaining the appropriate fluidity of biofilms during evolution [17]. Compared with other fatty acid desaturases, the activity of ∆9 desaturases was mostly involved in changing the membrane fluidity during temperature variation [71,72]. In fungi, low temperature improving the expression level of ∆15 desaturase causing omega-3 fatty acid production was proverbial [73,74]. However, according to Kosa’s study, the ∆15 desaturase activity showed opposite results in different species from a genus. Low temperatures (15 °C and 20 °C) enhance the activity of ∆15 desaturase, leading to an increase in ALA production in *Mucor flavus* CCM 8086 and VKM F-1003 [75]. Nevertheless, this study also found that the production of ALA was reduced due to the much weaker activity of ∆15 desaturase in *Mucor flavus* VKM-1097 cultured at 20 °C low temperature [75]. The ∆15 desaturase activity presents contrary regulation at low-temperature cultivation in different strains. In *Rhodotorula glutinis* YM25079, the expression level of ∆12 desaturase elevated five-fold when strain grown temperature from 25 °C to 15 °C [76]. Under low temperatures, the expression levels of all desaturase genes except ∆5 were increased in *Mortierella* sp. AGED [77]. In our study, the expression levels of ∆9 and ∆15 desaturases were up-regulated over 8- and 23-fold, respectively, in 10 °C growth conditions; the activity of ∆12 desaturase had no change.

According to the content of fatty acids, the deletion of *FgFAD12* led to the reduction of linoleic acid, 1.96% linolenic acid was detected. Based on unsaturated fatty acid biosynthesis of *F. graminearum* in KEGG, FGSG_05784 (*FgFAD12*) and FGSG_07890 (*FgFAD15*) both catalyze OA to LA in n-3 PUFAs biosynthesis pathway. *FgFAD15* plays the part of catalyzing OA to LA and then to ALA. It is possible that *FgFAD12* and *FgFAD15* have overlapping functions in catalyzing OA to LA in n-3 PUFA biosynthesis. It will be important to characterize the functional relationship of FgFad12 and FgFad15 and other desaturases in *F. graminearum*.

## 5. Conclusions

This study characterized *FgFAD12* in *F. graminearum* and revealed that *FgFAD12* was important in the biosynthesis of LA, vegetative growth and plant infections but had no effect on asexual and sexual reproduction. The deletion of *FgFAD12* led to an increased resistance to osmotic stress and reduced tolerance to oxidative stress. Different from *FgFAD9* and *FgFAD15*, the expression of *FgFAD12* was not affected by temperatures. These results contribute to revealing the mechanism of unsaturated fatty acids regulating *F. graminearum* pathogenesis and provide a basis for the synthesis of unsaturated fatty acids in other oil-producing filamentous fungi.

## Figures and Tables

**Figure 1 jof-10-00288-f001:**
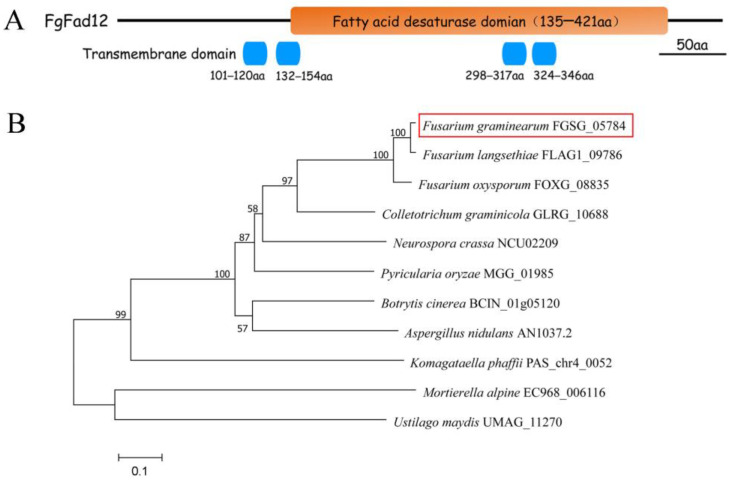
The identification of Fad12 ortholog in the genome of *F. graminearum.* (**A**) Domain structures of FgFad12. (**B**) The Neighbor-Joining tree of Fad12 orthologs. The phylogenetic tree was constructed using MEGA 7 with full-length sequences of proteins. FGSG_05784 was marked with red frame. Scale bar corresponds to 0.1 amino acid substitutions per site.

**Figure 2 jof-10-00288-f002:**
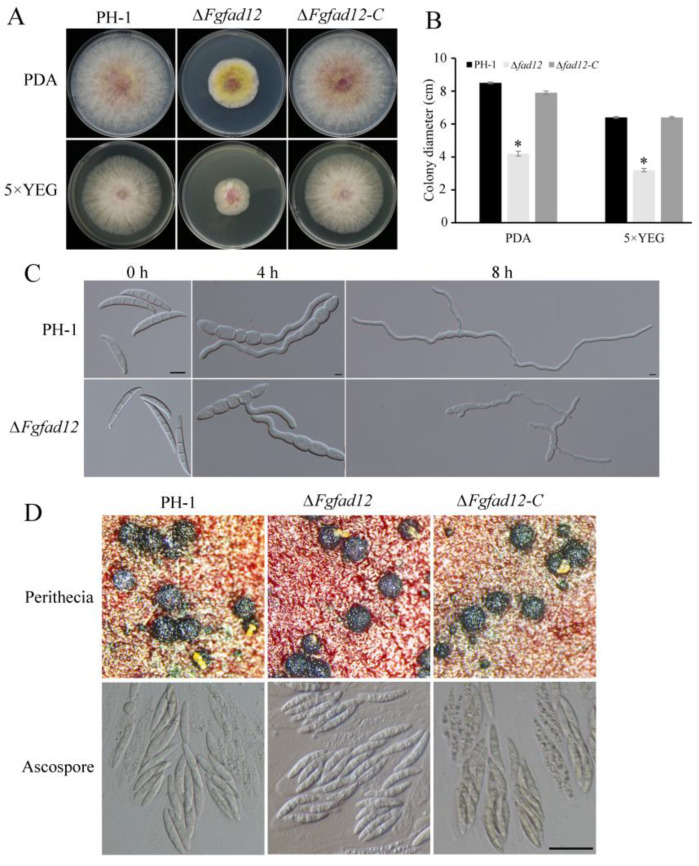
Phenotypes of ∆*Fgfad12* mutant in vegetative growth, conidial germination and sexual reproduction. (**A**) The wild-type PH-1, ∆*Fgfad12* mutant and ∆*Fgfad12-C* strains were cultured on PDA and 5 × YEG plates for 3.5 days. (**B**) Colony diameters of PH-1, ∆*Fgfad12* and ∆*Fgfad12-C* strains. Mean and standard deviation were calculated with data from three replicates. Asterisk indicates a statistically significant difference (*p* < 0.05) by Duncan’s multiple range test. (**C**) Conidial germination of PH-1, ∆*Fgfad12* mutant and ∆*Fgfad12-C* strains. Conidia were incubated in YEPD liquid medium, and images were taken at 0 h, 4 h and 8 h post-incubation. Bar = 20 μm. (**D**) Perithecia and ascospores on carrot agar cultures of PH-1, ∆*Fgfad12* mutant and ∆*Fgfad12-C* strains were examined after 10 days post-fertilization. Bar = 20 μm.

**Figure 3 jof-10-00288-f003:**
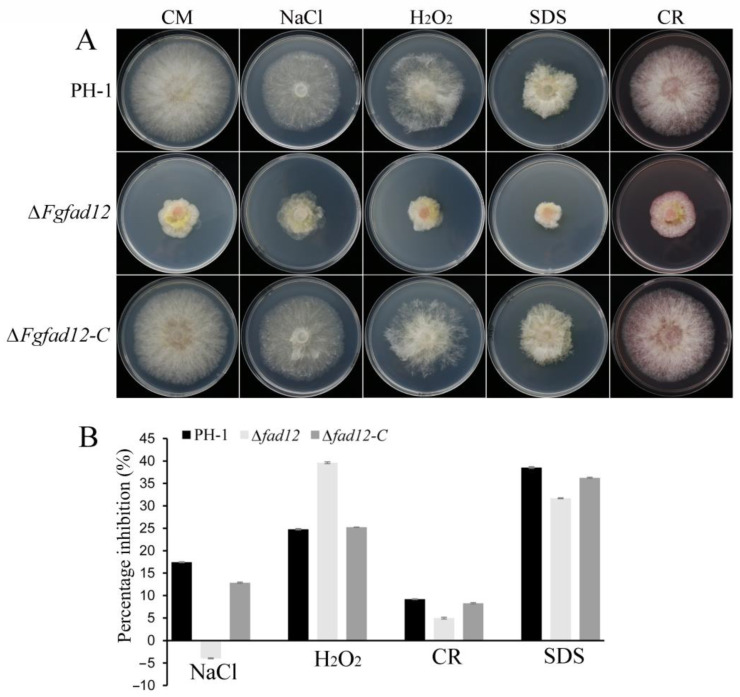
Phenotypes of ∆*Fgfad12* mutant in stress sensitivity. (**A**) PH-1, ∆*Fgfad12* and ∆*Fgfad12-C* strains were cultured on CM plates with or without 0.7 mol/L NaCl, 0.05% H_2_O_2_, 300 μg/mL Congo red and 0.01% SDS for 3 days. (**B**) Mycelial growth percentage inhibition of PH-1, ∆*Fgfad12* and ∆*Fgfad12-C* strains. Mean and deviation were calculated with data from three biological replicates.

**Figure 4 jof-10-00288-f004:**
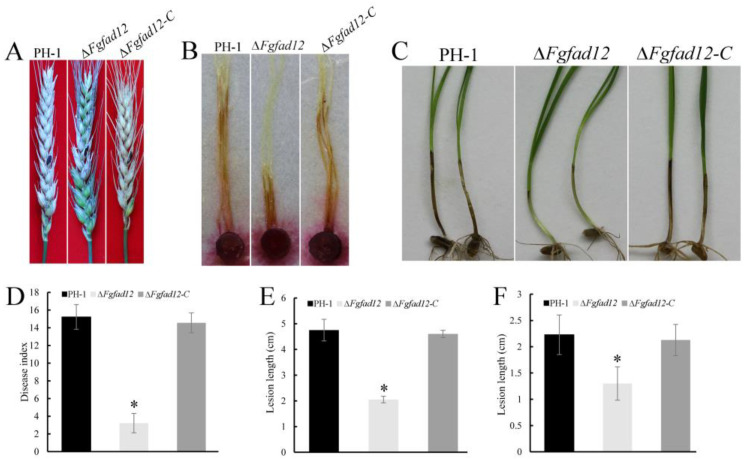
Defects in pathogenicity of ∆*Fgfad12* mutant. (**A**) Flowering wheat heads inoculated with PH-1, ∆*Fgfad12* and ∆*Fgfad12-C* strains were photographed 14 days post-inoculation (dpi). Black dots mark the inoculated spikelet. (**B**) Wheat coleoptiles were inoculated with the same strains and photographed at 14 dpi. (**C**) Corn silks were inoculated with culture blocks of the same set of strains, and photographs were taken at 6 dpi. (**D**) Disease index of the three strains determined at 14 dpi. More than 15 wheat heads were examined in each replicate. (**E**) Lesion length was examined in at least three biological replicates of each strain. (**F**) Lesion length was estimated with data from 20 coleoptiles of each strain. Asterisks indicate a statistically significant difference (*p* < 0.05) by Duncan’s multiple range test.

**Figure 5 jof-10-00288-f005:**
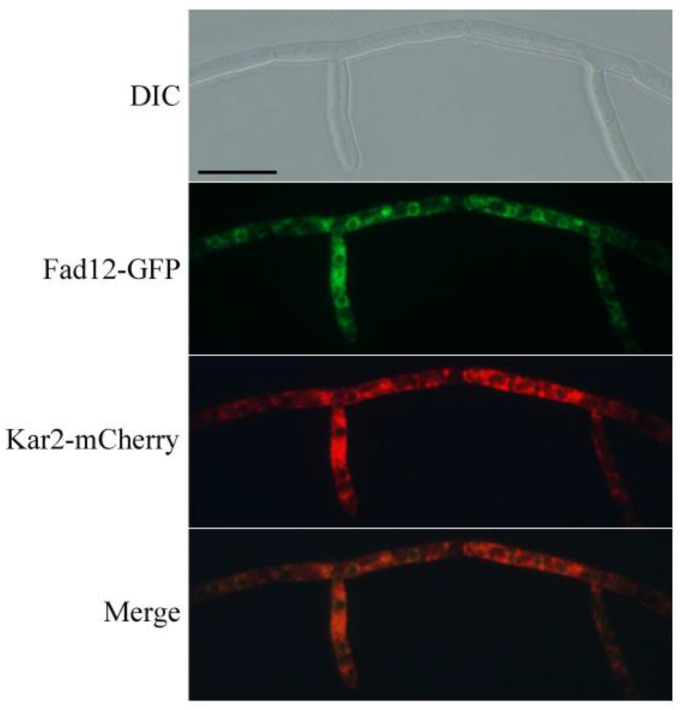
Subcellular localization of the FgFad12-GFP fusion protein. At 12 h, germlings of transformant expressing *FgFAD12*-GFP and *FgKAR2*-mCherry were examined by DIC and epifluorescence microscopy. Bar = 20 μm.

**Figure 6 jof-10-00288-f006:**
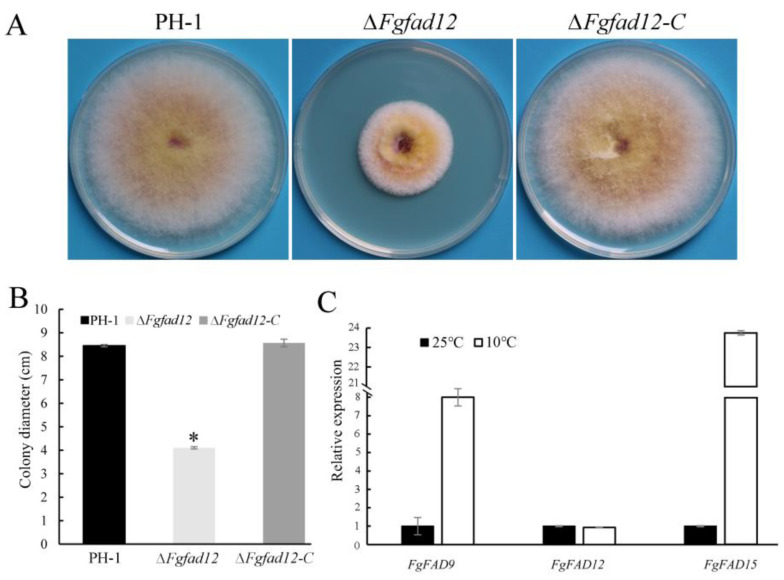
Assays for ∆*Fgfad12* mutant phenotype and desaturase genes expression at 10 °C. (**A**) The wild-type PH-1, ∆*Fgfad12* mutant and ∆*Fgfad12-C* strains were cultured on PDA plates for 10 days at 10 °C. (**B**) Colony diameters of PH-1, ∆*Fgfad12* and ∆*Fgfad12-C* strains. Colony diameters were estimated with three biological replicates of each strain. Asterisk indicates a statistically significant difference (*p* < 0.05) by Duncan’s multiple range test. (**C**) The expression levels of *FAD9*, *FAD12* and *FAD15* were assayed by qRT-PCR with PH-1 RNA isolated from 6-day vegetative hyphae cultured on PDA plates at 25 °C and 10 °C. The expression level of each gene in PH-1 was arbitrarily set to 1. Mean and deviation were calculated with data from three biological replicates.

**Figure 7 jof-10-00288-f007:**
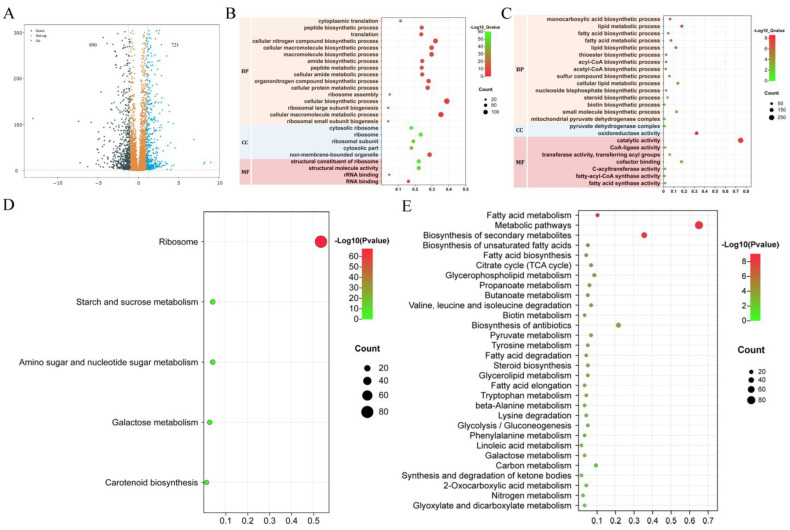
RNA-seq analysis of Fgfad12 mutant. (**A**) Volcano plot of significantly different genes in the Fgfad12 mutant. The numbers of up-regulated and down-regulated genes (log2FC > 1, *p* value < 0.05) with standard deviation were calculated with data from three biological replicates. (**B**) Gene Ontology (GO) terms enrichment analysis for significantly down-regulated genes. (**C**) Gene Ontology (GO) terms enrichment analysis for significantly up-regulated genes. BP, biological process; CC, cell component; MF, molecular function. (**D**) Top KEGG pathway terms for significantly down-regulated genes. (**E**) Top KEGG pathway terms for significantly up-regulated genes.

**Table 1 jof-10-00288-t001:** Wild type and transformants of *Fusarium graminearum* strains used in this study.

Strain	Brief Description	References
PH-1	Wild type	[46]
∆*Fgfad12*	*Fgfad12* deletion mutant of PH-1	This study
∆*FgFad12*-*C*	*Fgfad12/FgFAD12*-GFP transformant of ∆*Fgfad12*	This study
CL3	*FgFAD12*-GFP and *FgKAR2*-mCherry transformant of PH-1	This study

**Table 2 jof-10-00288-t002:** Defects of strains in growth, conidiation, pathogenicity and DON production.

Strain	Growth Rate ^a^ (mm/d)	Conidiation ^b^ (×10^6^/mL)	Disease Index ^c^	DON (ppm) ^d^
PH-1	12.1 ± 0.1 ^A^	1.5 ± 0.4 ^A^	15.2 ± 1.4 ^A^	718.5 ± 74.3 ^A^
∆*Fgfad12*	6.0 ± 0.2 ^B^	1.6 ± 0.3 ^B^	3.2 ± 1.1 ^B^	103.0 ± 27.7 ^B^
∆*FgFad12*-*C*	12.1 ± 0.1 ^A^	1.5 ± 0.2 ^A^	14.6 ±1.1 ^A^	709.5 ± 49.4 ^A^

Mean and standard deviations were determined from data of three biological replicates. Data were analyzed with the protected Fisher’s least significant difference (LSD) test. Different letters marked statistically significant differences (*p*< 0.05). ^A^ denoted no significant difference, while ^B^ denoted significant difference. ^a^ Growth rate was measured on PDA cultures. ^b^ Conidiation in 5 d CMC cultures. ^c^ Disease index was examined by the average number of symptomatic spikelets 14 dpi. Mean and standard deviation were calculated with three independent repeats. At least 15 wheat heads were examined in each replicate. ^d^ DON production was assayed with infected rice grain cultures.

**Table 3 jof-10-00288-t003:** Fatty acids content of PH-1 and ∆*FgFad12* strains.

Content of Fatty Acids (μg/mg)						
Strain	Capric Acid (C10:0)	Palmitic Acid (C16:0)	Stearic Acid (C18:0)	Oleic Acid (C18:1)	Linoleic Acid (C18:2)	Linolenic Acid (C18:3)
PH-1	0.13 ± 0.03	4.03 ± 0.11	1.20 ± 0.12	4.91 ± 0.17	14.66 ± 0.05	8.68 ± 0.13
∆*FgFad12*	0.11 ± 0.04	3.92 ± 0.49	1.42 ± 0.40	56.30 ± 0.67	0	1.23 ± 0.18

## Data Availability

RNA-seq data were deposited in the Genome Sequence Archive in the National Genomics Data Center, China National Center for Bioinformation/Beijing Institute of Genomics, Chinese Academy of Sciences (GSA: CRA012955) that are publicly accessible at https://ngdc.cncb.ac.cn/gsa (accessed on 8 October 2023).

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
