# Peer review of "FgFAD12 Regulates Vegetative Growth, Pathogenicity and Linoleic Acid Biosynthesis in Fusarium graminearum"

_jof, 2024, doi:10.3390/jof10040288_

Round 1

Reviewer 1 Report

The paper "FgFAD12 regulates vegetative growth, pathogenicity and linoleic acid biosynthesis in Fusarium graminearum" by Zhang and co-authors is well written and sheds ligth on some important roles of a fatty acid desaturase in the infection of F. graminearum. Few minor revisions, detailed below, are required before publication

Line 45: replace is with are

line 47 delete are the

line 87 C of cianobacterium in uppercase, s of symechocystis in lowercase

line 420 insert in before consistent

line 430 insert is before cultured

line 439 delete and before fluidity

line 440 add and before permeability

line 442 replace inoculated with after inoculation

line 452 replace increase with increased

line 453 add was before cultivated and replace in with at

line 457 replace improved with improving

line 472 replace were with was

Reviewer 2 Report

In this interesting work, in the phytopathogenic fungus Fusarium graminearum, the authors discovered and studied the role of the FAD12 gene, which encodes a fatty acid desaturase that catalyzes the conversion of oleic acid to linoleic acid. They found that a strain mutant for the FgFAD12 gene has defects in vegetative growth, conidia formation and pathogenesis, but reproduces normally sexually. There are no serious flaws in the work. The work is performed at a high methodological level. Below is a list of minor corrections and suggestions for further improvement of the work and increasing its scientific significance.

1.    The difference between conidiation and conidial germination needs to be further clarified. Conidia are spores of asexual reproduction, but it is said that although conidial germination disorders occur, the fungus has no problem with sexual reproduction, which may confuse the reader. 

2.    The introduction mentions three mycotoxins produced by F. graminearum: deoxynivallenol (DON), zearalenone (ZEN) and nivalenol (NIV). The paper shows that the mutant strain dFgFAD12 produces less DON than the wild-type strain. Do the authors have any data comparing ZEN and NIV content?

3.    Agents such as SDS and Congo red are used as stressors. It is necessary to explain how exactly they affect the cell as it is less obvious than for NaCl and H2O2

4.    The mutant strain is less susceptible to the effects of various stresses other than oxidative stress than the wild-type strain. Could this be explained, for example, by the fact that the strain has lower cell membrane fluidity due to the reduced PUFAs content?

5.    The expression of FgFAD9 and FgFAD15 genes in the wild-type strain is higher at 10 C than at 25 C. This is probably due to the need for increased membrane fluidity in response to lower temperature. Could a similar comparison be performed for the expression of FgFAD9 and FgFAD15 in the dFgFAD12 mutant at 10 C VS 25 C?

6.    The findings suggest that perhaps the functions of FgFAD12 and FgFAD15 overlap and that FgFAD15 may be responsible for the presence of linolenic acid in the mutant strain. But why then is linoleic acid, which is a putative intermediate, not detected in it even in trace amounts? Is an alternative pathway of linolenic acid biosynthesis possible, or is linoleic acid consumed much faster than it is formed, having a very low steady-state concentration?

7.    Transcriptome data show that expression of genes encoding enzymes for ergosterol biosynthesis is elevated in the mutant strain, which may be a response to reduced PUFAs content in membranes. Can the mutant strain and the wild-type strain be further analyzed and compared for ergosterol content?

Round 2

Reviewer 2 Report

The authors have substantially improved the manuscript in accordance with all the comments. I recommend that the revised manuscript be accepted for publication in the journal.

The manuscript has been corrected in accordance with all comments.